# Screening Effective Antifungal Substances from the Bark and Leaves of *Zanthoxylum avicennae* by the Bioactivity-Guided Isolation Method

**DOI:** 10.3390/molecules24234207

**Published:** 2019-11-20

**Authors:** Yongtong Xiong, Guan Huang, Zongli Yao, China Zhao, Xiang Zhu, Qinglai Wu, Xudong Zhou, Junkai Li

**Affiliations:** 1School of Agriculture, Yangtze University, Jingzhou 434025, China; Xiongyongtong0124@163.com (Y.X.); Huang0506@163.com (G.H.); y806032526@163.com (Z.Y.); zhaochina0623@163.com (C.Z.); cjdxnxyzx@sina.com (X.Z.); 2Institute of Pesticides, Yangtze University, Jingzhou 434025, China; 3TCM and Ethnomedicine Innovation & Development Laboratory, School of Pharmacy, Hunan University of Chinese Medicine, Changsha 410208, China

**Keywords:** *Zanthoxylum avicennae*, bioactivity-guided isolation method, antifungal activity, lead compound, fungicide

## Abstract

To find good antifungal substances by the bioactivity-guided isolation method, we tracked down the effective antifungal substances in the bark and leaves of *Zanthoxylum avicennae*, and isolated three antifungal compounds **1**, **2**, and **3.** The structures were identified as xanthyletin, luvangetin, and avicennin by ^1^H-NMR, ^13^C-NMR, and HRMS spectra. Particularly, compound **2** had several isomers, and the ^1^H-NMR spectra of **2** in different solvents showed a significant difference. To determine the stereo structure of **2,** a single crystal was prepared and identified by X-ray diffraction as Luvangetin. Moreover, the difference of ^1^H-NMR data of **2** between in solvent dimethyl sulfoxide-*d_6_* (DMSO-*d_6_*) and deuterated chloroform (CDCl_3_), and other reported isomers were discussed for the first time. The bioassay results indicated that the three compounds **1**, **2**, and **3** displayed low to high antifungal activities against tested phytopathogenic fungi. In particular, all compounds **1**, **2**, and **3** showed excellent antifungal activities against *Pyricularia oryzae* and *Z. avicennae*, with the values of half maximal effective concentration (EC_50_) ranging from 31 to 61 mg/L, and compound **3** was also identified as a more potent inhibitor against *Fusaium graminearum* (EC_50_ = 43.26 ± 1.76 mg/L) compared with fungicide PCA (phenazine-1-carboxylic acid) (EC_50_ = 52.34 ± 1.53 mg/L). The results revealed that compounds **1, 2,** and **3** were the main antifungal substances of *Z. avicennae*, and can be used as lead compounds of a fungicide, which has good development value and prospect.

## 1. Introduction

*Zanthoxylum avicennae* belongs to *Rutaceae*, and is mainly distributed around southern coastal area of China and parts of Southeast Asia, such as Hainan, Fujian, Guangxi, the Philippines, and Vietnam [1]. *Z. avicennae* has often been used to treat and relieve many illnesses, such as multi-phlegm, rheumatism, sore throat, jaundice, insect and snake bites, vomiting and diarrhea, repel *Ascaris*, and digestive system diseases in clinical and in traditional Chinese medicine [2,3,4,5,6,7,8]. Therefore, *Z. avicennae* deserves great research value and the bioactive substances from *Z. avicennae* have attracted researchers’ attention. Further studies have shown that *Z. avicennae* mainly contains the compounds of triterpenes, steroids, alkaloids, amides, lignans, coumarins, flavonoids, volatile oils, and fatty acids [6,9,10,11,12], and many compounds have been proven to have various bioactivities, such as anti-cancer, antibacterial [13], antimildew activities [14], inhibitory activities against *Plasmodium vivax* [15], and 1,1-diphenyl-2-picrylhydrazyl (DPPH) free radical-scavenging activity [16].

In China, *Z. avicennae* is distributed abundantly; however, research on the antifungal activity of *Z. avicennae* is seldom reported. In order to develop and utilize this plant resource to the best advantage, and expand the antifungal application of *Z. avicennae*, we used the bioassay-guided isolation method to focus on exploring the antifungal compounds of *Z. avicennae*. The results provide a research basis and helpful clues for the discovery of novel antimicrobial agents.

## 2. Results and Discussion

In this study, 28.0 g of bark crude extract were obtained from 120.0 g of bark, and 45.0 g of leave crude extract were obtained from 170.0 g of leaves of *Z. avicennae*. The extraction yields were 23.3% and 26.5%, respectively. Eight fractions (**A–H**) were isolated from the bark crude of *Z. avicennae* and four fractions (**I–L**) were isolated from the leave crude. By the bioassay-guided isolation method, three bioactive compounds **1, 2,** and **3** were isolated from the fractions, and all compounds were identified by ^1^H-NMR, ^13^C-NMR, and HRMS spectra (see details in Appendix A). Detailed analysis indicated that compounds **1–3** were xanthyletin [17,18], luvangetin [19,20], and avicennin [21,22], respectively, as shown in Figure 1.

However, we found that the NMR spectra of three compounds in solvent DMSO-*d*_6_ and deuterated chloroform (CDCl_3_) showed apparent differences. Particularly, the difference of the ^1^H-NMR spectra of compounds **1** and **2** were especially remarkable. In this research, the ^1^H and ^13^C-NMR spectra of compounds **1**, **2** and **3** in solvent DMSO-*d*_6_, and the differences of ^1^H-NMR spectra in DMSO-*d*_6_ and CDCl_3_ were reported for the first time, as shown in Table 1. Moreover, compound **2** had several natural isomers, such as luvangetin [19,20], alloxanthoxyletin [20], and xanthoxyletin [17,23], shown in Figure 2, and the ^1^H-NMR spectra of these compounds were very similar. Particularly, the ^1^H-NMR spectrum of compound **2** in DMSO-*d*_6_ was almost consistent with the reported spectrum of xanthoxyletin in CDCl_3_ [17]. To determine the stereo structure of **2**, a single crystal was prepared by slow evaporated in ethyl acetate at room temperature, and it was confirmed as luvangetin by X-ray diffraction, as shown in Figure 3.

According to the bioassay-guided isolation method, first, the crude extracts of *Z. avicennae* bark and leaves were screened for antifungal activities against six phytopathogenic fungi, such as *Rhizoctonia solani, Fusaium graminearum*, *Altemaria solani*, *Fusarium oxysporum*, *Sclerotinia sclerotiorum*, and *Pyricularia oryzae*, at the concentration of 500 mg/L, as shown in Table 2 Considering the crude extract contained a large number of other substances, such as polysaccharides and proteins, and the contents of bioactive constituents were very low, the concentration of 500 mg/L was selected for screening to not miss the active ingredients. The results indicated that the crude extracts of *Z. avicennae* bark and leaves have certain antifungal activities against most tested phytopathogenic fungi. In particular, both the crude extracts of *Z. avicennae* bark and leaves showed moderate antifungal activities against *R. solani*. The inhibition rates were 61.89 ± 2.81% and 56.07 ± 1.76%, respectively. The crude extracts of *Z. avicennae* bark also had moderate to high antifungal activities against *S. sclerotiorum* and *P. oryae*; the inhibition rates were 56.05 ± 1.25% and 93.5 ± 1.28%, respectively.

Second, the crude extracts of *Z. avicennae* bark and leaves were separated by silica gel (200–300 mesh) column chromatography to obtain eight fractions (**A-H**) and four fractions (**J-L**), respectively. All fractions were tested for their antifungal activities against the same six phytopathogenic fungi at the concentration of 50 mg/L, as shown in Table 3. The results showed that the fractions of *Z. avicennae* leaves had no antifungal activities to most phytopathogenic fungi. Only fraction **I** demonstrated activity against *P. oryae* and *F. oxysporum* with inhibition rates of 65.02 ± 2.58% and 59.52 ± 2.37%, and fraction **J** against *P. oryae* with an inhibition rate of 66.46 ± 2.43%. The fractions **E–H** of *Z. avicennae* bark also had no or very low antifungal activities to the tested phytopathogenic fungi, but the fractions **A–D** showed low to high antifungal activities. Particularly, fraction **A** showed the best antifungal activity against *P. oryae* with an inhibition rate of 96.05 ± 0.81%. Fraction **B** had good antifungal activities against *R.solani* and *P. oryae* with inhibition rates of 86.18 ± 0.52% and 73.03 ± 2.62%, and fraction **C** indicated high antifungal activities against *F.graminearum* and *P. oryae* with inhibition rates of 87.89 ± 0.82% and 82.78 ± 0.94%. The bioassay-guided results revealed that the antifungal substances mainly existed in fractions **A**, **B** and **C**.

Finally, fractions **A**, **B**, and **C** were separated and purified by silica gel (200–300 mesh) column chromatography again, and gained three purified compounds **1, 2,** and **3**, which were identified as xanthyletin, luvangetin, and avicennin by ^1^H-NMR, ^13^C-NMR, and HRMS spectra. The results of the antifungal activities indicated that compounds xanthyletin, luvangetin, and avicennin also had good antifungal activities against *F.graminearum*, *R.solani*, and *P. oryae*, just like fractions **A**, **B**, and **C,** shown in Table 4. So, the EC_50_ values of xanthyletin against *P. oryae*, luvangetin against *R. solani,* and avicennin against *F. graminearum* and *P. oryae* were determined, as listed in Table 5. The results showed that the antifungal activities of xanthyletin (**1**) and lcuvangetin (**2**) against *P. oryae* were slightly lower than PCA (29.30 ± 1.89 mg/L), with EC_50_ values of 31.56 ± 1.86 and 35.89 ± 1.64 mg/L, respectively. Noteworthily, avicennin (**3**) showed higher antifungal activity against *F. graminearum*, with EC_50_ values of 43.26 ± 1.76 mg/L, than PCA (52.34 ± 1.53 mg/L). The results revealed that xanthyletin, luvangetin, and avicennin are the main antifungal substances of *Z. avicennae*.

## 3. Materials and Methods

Chemicals and solvents were purchased from commercial suppliers in China and were used without further purification. Solvents and reagents were abbreviated as follows: Menthol (CH_3_OH), chloroform (CHCl_3_), petroleum ether, dichloromethane (DCM), ethyl acetate (EtOAc), and sodium sulfate (Na_2_SO_4_). All fungi were obtained from the School of Agricultural, Yangtze University (CN). The melting points were determined on a WRR melting point apparatus (Shanghai Jingke Industrial Co. Ltd., Shanghai, China) and modified. Thin-layer chromatography (TLC) was performed on silica gel 60 F254 (Qingdao Marine Chemical Ltd., Qingdao, China). Column chromatography (CC) was performed over silica gel (200–300 mesh, Qingdao Marine Chemical Ltd., Qingdao, China). ^1^H and ^13^C-NMR spectrum were recorded in CDCl_3_ or DMSO-*d_6_* solution on a Bruker 400 MHz spectrometer (Bruker Co., Fällanden, Switzerland), using tetramethyl silane (TMS) as an internal standard, and chemical shift values (*δ*) are given in parts per million (ppm). The following abbreviations were used to designate chemical shift multiplicities: *s* = singlet, *d* = doublet, *t* = triplet, *q* = quartet, *m* = multiple. MS data were obtained using an APEX IV Fourier-Transform Mass Spectrometry (Bruker Daltonics, Billerica, MA, USA).

### 3.1. Plant Material

The bark and leaves of *Z. avicennae* were collected from Minlishan Mountain, Hezhou, China, and dried at room temperature (25 °C), then pulverized by plant grinder, and kept sealed in a dark place for later use.

### 3.2. Extraction and Isolation

In total, 120.0 g of bark powder were extracted with 400 mL of MeOH at room temperature for 2 days, and then the extract liquid was filtered and collected. The filter cake was repeatedly treated in the same way another two times, and combined three filtrates. In total, 170.0 g of leave powder of *Z. avicennae* were treated in the same way. The extracted solution was freeze-dried at −50 °C to obtain the crude extract, and the extraction yield was calculated. The crude extracts of *Z. avicennae* leaves and bark were 45.0 and 28.0 g, respectively, and the extraction yields were 26.5% and 23.3%, respectively.

The crude extract of *Z. avicennae* bark (25.0 g) was applied to silica gel (200–300 mesh) column chromatography, and eluted with a petroleum ether (PE)/ethyl acetate (EA) and ethyl acetate (EA)/methanol (MeOH) gradient system (PE/EA = 10:0, PE/EA = 8:1, PE/EA = 6:1, PE/EA = 4:1, PE/EA = 2:1, EA/MeOH = 1:1, EA/MeOH = 0:1) to give eight fractions (**A–H**). The crude extract of *Z. avicennae* leaves (40.0 g) was treated by a similar procedure to give four fractions (**I–L**). By the bioassay-guided isolation method, all fractions were tested for their antifungal activities and highly active ones were screened (**A**, **B**, and **C**). Then, fraction **A** was separated and purified by silica gel (300–400 mesh) column chromatography, which was eluted with a CHCl_3_/MeOH (90:100–80:20) gradient system to get purified compound **1**. Fraction **B** was recrystallized three times by chloroform at −4 °C to give purified compound **2**. Fraction **C** was separated and purified by silica gel (300–400 mesh) column chromatography (eluted with CHCl_3_/MeOH (100:0–80:20) gradient system) and Waters Breeze™ 2 HPLC (80% MeOH/H_2_O, flow rate 20 mL/min) to obtain purified compound **3**.

### 3.3. Primary Fungicidal activities

The primary fungicidal activities of all crude extracts, fractions, and purified compounds were tested against six phytopathogenic fungi (*Rhizoctonia solani, Fusaium graminearum*, *Altemaria solani*, *Fusarium oxysporum*, *Sclerotinia sclerotiorum*, and *Pyricularia oryzae*) by the mycelium growth rate isolation method [23], in which the concentration of crude extracts was 500 mg/L and the concentration of fractions and compounds was 50 mg/L. The commercial fungicide Shenqinbactin (PCA) was assessed as positive controls at the concentration of 50 mg/L. Each treatment was repeated three times. When the mycelia of CK grew to three quarters of the area of the diameter, it was measured by the cross isolation method [24,25], and the inhibition ratio was calculated by Equation (1):Relative inhibition ratio (%) = [(CK − PT) / (CK − 6 mm)] × 100%(1)
where CK is the extended diameter of the circle mycelium during the negative control and PT is the extended diameter of the circle mycelium during experiment.

### 3.4. EC_50_ Value of Fungicidal Activities

The EC_50_ values of the purified compounds obtained by the bioassay-guided isolation method were determined, and the final concentrations of all compounds in medium were 200, 150, 100, 50, 25, and 12.5 mg/L. The isolation method of inoculating phytopathogenic fungi cakes and circle mycelium measurement was equal to the isolation method for testing the primary antifungal activities. All fungicidal activities were evaluated by statistical analysis. All statistical analysis was performed using EXCEL 2010, software (Microsoft, Redmond, WA, USA). The log dose–response curves allowed determination of the EC_50_ for the fungi bioassay according to probit analysis. The 95% confidence limits for the range of EC_50_ values were determined by the least-square regression analysis of the relative growth rate (% control) against the logarithm of the compound concentration.

## 4. Conclusions

*Z. avicennae*, which has extensive pharmacological activities, was used as experimental material. The crude extracts of *Z. avicennae* bark and leaves were extracted by a cold soaking isolation method. By the bioassay-guided isolation method, we found that antifungal substances mainly exist in the bark of *Z. avicennae*, and three antifungal fractions **A, B,** and **C** were identified. Further separation and purification of fractions **A, B,** and **C** led to three purified compounds **1–3**, which were identified as xanthyletin, luvangetin, and avicennin by ^1^H-NMR, ^13^C-NMR, and HRMS spectra. The bioactivities of compounds **1–3** indicated that the three compounds had excellent antifungal activities against *F. graminearum*, *R. solani*, and *P. oryae.* In particular, xanthyletin and avicennin showed slightly lower antifungal activities against *P. oryae* than PCA (29.30 ± 1.89 mg/L), with EC_50_ values of 31.56 ± 1.86 and 35.89 ± 1.64 mg/L, respectively. Avicennin showed higher antifungal activities against *F. graminearum* than PCA (52.34 ± 1.53 mg/L), with an EC_50_ value of 43.26 ± 1.76 mg/L. So, compounds **1–3** can be confirmed as the main antifungal substances of *Z. avicennae*, and can be used as lead compounds of a fungicide, which has good development value and prospect.

## Figures and Tables

**Figure 1 molecules-24-04207-f001:**
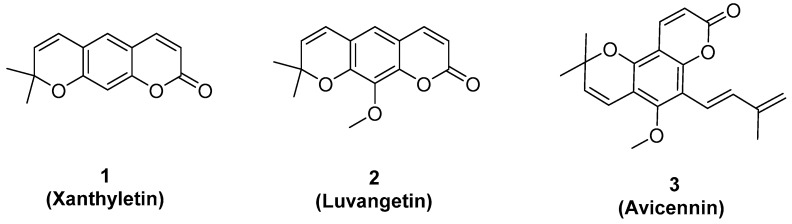
The structures of compounds **1**–**3**.

**Figure 2 molecules-24-04207-f002:**
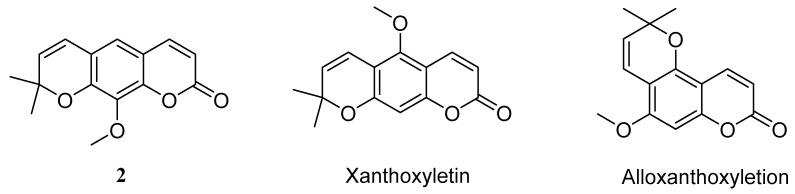
The natural isomers of **2**.

**Figure 3 molecules-24-04207-f003:**
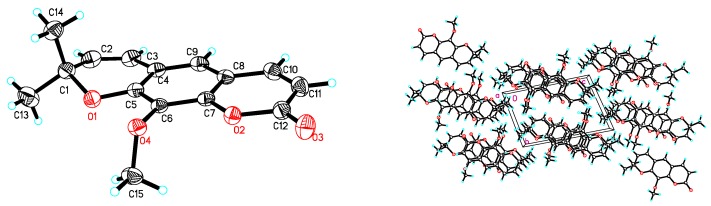
The X-ray crystal structure and crystal packing of **2.**

**Table 1 molecules-24-04207-t001:** ^1^H-NMR data (*J*, Hz) of **2** in different deuterium solvents (400 MHz) compared with the isomers reported by the literature.

Location	2 (DMSO-*d*_6_)	2 (CDCl_3_)	2 (In Ref.)	Alloxanthoxyletin	Xanthoxyletin
**C_2_-H**	**7.92 (d, 9.6, 1 H)**	**7.59 (d, 9.6, 1 H)**	**7.57 (d, 9.6, 1 H)**	**7.96 (d, 10.0, 1 H)**	**7.85 (d, 10.0, 1 H)**
C_3_-H	7.18 (s, 1H)	6.85 (s, 1H)	6.83 (s, 1H)	6.36 (s, 1H)	6.57 (s, 1H)
C_5_-H	6.48 (d, 10.0, 1H)	6.35 (d, 10.0, 1H)	6.33 (d, 10.0, 1 H)	6.62 (d, 10.0, 1H)	6.58 (d, 10.0, 1H)
C_7_-H	6.29 (d, 9.6, 1H)	6.24 (d, 9.6, 1H)	6.24 (d, 9.6 Hz, 1H)	6.16(d, 10.0, 1H)	6.21(d, 10.0, 1H)
C_8_-H	5.88 (d, 10.0, 1H)	5.72 (d, 10.0, 1H)	5.71 (d, 10.0, 1 H)	5.55 (d, 10.0, 1H)	5.71(d, 10.0, 1H)
C_1′_-H	3.86 (s, 3H)	3.99 (s, 3H)	3.98 (s, 3H)	3.87 (s, 3H)	3.87(s, 3H)
C_2′_-H, C_3′_-H	1.46 (s, 6H)	1.53 (s, 6H)	1.51 (s, 6H)	1.47 (s, 6H)	1.47(s, 6H)

**Table 2 molecules-24-04207-t002:** The in vitro antifungal activities of crude extract from bark and leaves of *Zanthoxylum avicennae* (*Z. avicennae*).

Crude Extract *	Inhibition Rate %					
	*R. solani*	*F. graminearum*	*A. solani*	*F. oxysporum*	*S. sclerotiorum*	*P. oryae*
bark	61.89 ± 2.81	19.70 ± 1.47	2.14 ± 0.82	19.44 ± 1.56	56.05 ± 1.25	93.5 ± 1.28
leaves	56.07 ± 1.76	0.00 ± 0.00	0.00 ± 0.00	45.15 ± 0.00	27.70 ± 0.94.	15.50 ± 0.82

^*^ at the concentration of 500 mg/L.

**Table 3 molecules-24-04207-t003:** The in vitro antifungal activities of the fractions isolated from *Z. avicennae* bark and leaves.

Part	Fractions *	Inhibition Rate %					
		*R. solani*	*F. graminearum*	*A. solani*	*F. oxysporum*	*S. sclerotiorum*	*P. oryae*
**bark**	A	51.32 ± 0.94	22.05 ± 0.82	54.10 ± 0.91	24.26 ± 1.25	50.98 ± 6.13	96.05 ± 0.81
	B	86.18 ± 0.52	18.90 ± 0.47	29.51 ± 1.25	33.82 ± 0.82	43.14 ± 3.86	73.03 ± 2.62
	C	66.81 ± 1.76	87.89 ± 0.82	13.63 ± 2.93	16.45 ± 0.81	44.14 ± 0.49	82.78 ± 0.97
	D	51.33 ± 1.70	26.77 ± 2.87	0	14.71 ± 0.45	0	46.71 ± 2.45
	E	0	0	8.51 ± 3.74	7.94 ± 1.25	0	30.57 ± 1.71
	F	0	10.00 ± 0.00	2.13 ± 0.82	0.79 ± 0.43	0	0
	G	0	12.31 ± 2.94	2.14 ± 0.81	6.35 ± 1.73	0	14.65 ± 1.79
	H	0	0	3.55 ± 3.77	0	28.41 ± 1.42	13.38 ± 0.86
**leaves**	I	25.83 ± 1.53	0	0	65.02 ± 2.58	27.59 ± 1.76	59.52 ± 2.37
	J	34.52 ± 1.86	0	0	8.22 ± 1.72	13.73 ± 0.85	66.46 ± 2.43
	K	0	0	6.35 ± 1.27	0	0	0
	L	0	0	0	5.38 ± 0.82	13.71 ± 1.27	8.28 ± 0.81

^*^ at the concentration of 50 mg/L.

**Table 4 molecules-24-04207-t004:** The in vitro antifungal activities of compounds **1**–**3**.

Compound	Inhibition Rate %					
	*R. solani*	*F. graminearum*	*A. solani*	*F. oxysporum*	*S. sclerotiorum*	*P. oryae*
**1**	57.62 ± 1.36	31.80 ± 2.15	63.83 ± 0.56	26.28 ± 0.93	63.03 ± 0.65	98.32 ± 1.67
**2**	88.23 ± 1.05	16.86 ± 1.57	33.43 ± 0.85	38.61 ± 1.93	45.25 ± 2.72	75.87 ± 0.86
**3**	69.38 ± 2.32	89.03 ± 1.74	13.36 ± 2.65	17.67 ± 1.56	52.35 ± 1.78	86.06 ± 1.81
PCA	97.65 ± 1.26	48.78 ± 1.64	74.65 ± 0.82	52.08 ± 1.23	87.86 ± 0.52	87.59 ± 0.65

^*^ at the concentration of 50 mg/L.

**Table 5 molecules-24-04207-t005:** The half maximal effective concentration (EC_50_) values (mg/L) of compounds **1–3.**

Compounds	*R. solani*	*F. graminearum*	*P. oryae*
**1**	49.10 ± 1.03	117.21 ± 0.93	31.56 ± 1.86
**2**	80.18 ± 2.65	134.26 ± 1.19	35.89 ± 1.64
**3**	40.58 ± 2.59	43.26 ± 1.76	61.62 ± 2.12
**PCA**	23.21 ± 2.23	52.34 ± 1.53	29.30 ± 1.89

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
