# Peer review of "Screening Effective Antifungal Substances from the Bark and Leaves of Zanthoxylum avicennae by the Bioactivity-Guided Isolation Method"

_molecules, 2019, doi:10.3390/molecules24234207_

Round 1

Reviewer 1 Report

This manuscript mentions the isolation, structures and biological activities of three known compounds. 

The isolation process was general, and the reason why these three  compounds were produced by this organism was not revealed, even three compounds were known. These three compounds had no remarkable biological activities.

I think this manuscript is not attracted to the readers of "Molecules".

Author Response

Responses to reviewers' comments:

Reviewer 1:

The isolation process was general, and the reason why these three compounds were produced by this organism was not revealed, even three compounds were known. These three compounds had no remarkable biological activities.

Response: Thank you for your good suggestions. Our interest focus on discovering bioactive contituents from natural products, rather than only new compounds. We have looked up lots of literature on your prestigious Molecules and find the content of the article is consistent with Molecules. And we found a very relevant article published in Molecules (Bioactivity-Guided Isolation of Antimicrobial and Antioxidant Metabolites from the Mushroom Tapinella atrotomentosa, Molecules, 2018, 23, 1082). So we sincerely hope that the revised manuscript will be worthy of publication in Molecules.

Reviewer 2 Report

The article was accept after Major Revisions

The article “Screening the Effective Antifungal Substances from the Bark and Leaves of Zanthoxylum avicennae by Bioactivity-guided Isolation Method” seems to have been prepared carelessly..

It must be subject to major revisions.

Introduction section:

It is necessary to expand the background, search for more recent literature on the subject.

Results and Discussion section:

Make the tables clearer and correct the graphic errors in the underlines. Line 67: The spectra of coumpounds 1,2 and 3 are not shown in tab 1 as it was written in the text. Why was the concentration of 500 mg / L t used for the crude extract? The authors should explain better if it corresponds to the EC50. Line 87-88: Invert the corresponding values respectively for bark and leaves. Is the 50 mg / L concentration used for all fractions the value of the EC50? Have the authors evaluated antifungal activities also against some resistant phytopathogenic fungi? The description of the results does not correspond to what is indicated in tab 3 for the different fractions. For eg: Has the fraction K of the leaves of Z. avicennae had a zero antifungal activity against P. oryae?!? Tab 5 is unclear. The EC50 of compounds found are superior to those of the PCA! was indicated instead slightly lower.. Did the authors statistically evaluate fungicidal activities? It would be appropriate to indicate whether statistical analysis has been performed

Materials and methods section:

It is necessary to better describe the protocol used for determined the EC50 value of Fungicidal activities. In the materials and methods there is no paragraph with statistical analysis performed

Conclusion section:

The conclusion should be broadened. Is it possible to do more speculation on the use of these substances for the treatment of fungal infections? In particular against pathogens resistant to currently known antifungal substances. Is it possible to make hypotheses on the mechanism of action of these substances on the basis of the present literature?

Numerous print errors should be corrected eg:

Line 185 and 186  luvangetin … lowercase

In figure 2 was written alloxanthoxyletion (line 77)

Furthermore, poor and not recent bibliography.

Author Response

Reviewer 2:

The article “Screening the Effective Antifungal Substances from the Bark and Leaves of Zanthoxylum avicennae by Bioactivity-guided Isolation Method” seems to have been prepared carelessly.

Response: Thank you for your good suggestions. We polished and reedited the article carefully.

Introduction section:

It is necessary to expand the background, search for more recent literature on the subject.

Response: Thank you for your good suggestions. The recent related reference has been added. Please see reference 2-4.

Results and Discussion section:

Make the tables clearer and correct the graphic errors in the underlines. Line 67: The spectra of coumpounds 1,2 and 3 are not shown in tab 1 as it was written in the text.

Response: Thank you for your good suggestions. The table 1 has been reedited. Please see Table 1.

Why was the concentration of 500 mg / L t used for the crude extract? The authors should explain better if it corresponds to the EC50.

Response: Thank you for your good suggestions. Because the crude extract contained a large number of other substances such as polysaccharides and proteins, and the contents of bioactive constituents were very low, so the concentration of 500 mg / L is selected for screening so as not to miss the active ingredients.-----This is not directly related to the latter EC50 of those pure compounds.

Line 87-88: Invert the corresponding values respectively for bark and leaves.

Response: Thank you for your good suggestions. We have made some revisions in the MS which may cause misunderstanding. 

Is the 50 mg / L concentration used for all fractions the value of the EC50?

Response: Thank you for your good suggestions. The concentration of 50 mg/L was use as a screening concentration for each pure compound, not EC50. The values of EC50 were calculated after detecting the fungicidal activity of 5 gradient concentrations.

Have the authors evaluated antifungal activities also against some resistant phytopathogenic fungi?

Response: Thank you for your good suggestions. We only tested plant pathogenic fungi, but did not test some resistant phytopathogenic fungi.

The description of the results does not correspond to what is indicated in tab 3 for the different fractions. For eg: Has the fraction K of the leaves of Z. avicennae had a zero antifungal activity against P. oryae?!?

Response: Thank you for your good suggestions. We are very sorry for the misplacement of the table data after my original manuscript was retypesetting by the manuscript system. We apologize for the trouble that caused you. The fraction K of the leaves of Z. avicennae has no effect on P. oryae.

Tab 5 is unclear. The EC50 of compounds found are superior to those of the PCA! was indicated instead slightly lower.

Response: Thank you for your good suggestions. EC50 is related to the inhibitory concentration, and the higher the value, the worse the efficacy. Therefore, the results in the table showed that most of the efficacy of the three compounds against the three tested pathogens was lower than PCA, and only Avicennin (3) against F. graminearum was slightly higher than PCA.

Did the authors statistically evaluate fungicidal activities? It would be appropriate to indicate whether statistical analysis has been performed  

Response: Thank you for your good suggestions. We have added the value±SD of three individual observations for the tested isolates and all fungicidal activities were evaluated by statistical analysis

Materials and methods section:

It is necessary to better describe the protocol used for determined the EC50 value of Fungicidal activities. In the materials and methods there is no paragraph with statistical analysis performed

Response: Thank you for your good suggestions. The method for determining the EC50 value of fungicidal activities were described in the part of materials and methods and these methods were based on the reference.

Conclusion section:

The conclusion should be broadened. Is it possible to do more speculation on the use of these substances for the treatment of fungal infections? In particular against pathogens resistant to currently known antifungal substances. Is it possible to make hypotheses on the mechanism of action of these substances on the basis of the present literature?

 Response: Thank you for your good suggestions. We resummarize and reedited the conclusion. Subsequent work will work on the mechanism of action of these substances.

Numerous print errors should be corrected eg:Line 185 and 186  luvangetin … lowercase. In figure 2 was written alloxanthoxyletion (line 77)

 Response: Thank you for your good suggestions. The corresponding revision has been done.

Furthermore, poor and not recent bibliography.

Response: Thank you for your good suggestions. The recent related reference has been added. Please see reference.

Round 2

Reviewer 1 Report

This manuscript does not seem to be suitable for this journal because the molecular structures are not  novel and biological activity of these compounds are not potent.

Author Response

1.This manuscript does not seem to be suitable for this journal because the molecular structures are not novel and biological activity of these compounds are not potent.

Response: Thank you for your suggestions. In this research, the purpose of the bioactivity-guided isolation method was to find antifungal natural products quickly,and utilize these antifungal natural products as lead compounds for structural modification, which can develop novel efficient low-toxicity fungicide based on the natural skeleton in the future work. Furthermore, the antifungal natural products in this research showed good antifungal activities against Pyricularia oryzae, Z. avicennae and Fusaium graminearum, and the antifungal activities were equal to the level of the commercial fungicide PCA (Phenazino-1-carboxylic acid ; Shenqinmycin). Thus it was meaningful has good research value. So we sincerely hope that the revised manuscript will be worthy of publication in Molecules.

Reviewer 2 Report

Introduction section

ref 12 (Sandjo, L. P., V. Kuete, R. S. Tchangna, T. Efferth and B. T. Ngadjui. 2014. Cytotoxic benzophenanthridine 221 and furoquinoline alkaloids from Zarrthoxylum buesgenii (Rutaceae). Chem. Cent. J. 8:1-5.) is not present in the introduction.

Results and Discussion section

Insert an explanation in the text to choose the concentrations of 500 mg / L (used for the crude extract) and 50 mg / L (as a screening concentration for each pure compound).

Table 3 remains unclear and the text still indicates that “fractions K against P. oryae showed moderate antifungal activities …”

I know perfectly the meaning of the EC50, so it seems to me that the explanation of the results of tab 5 is not correctly expressed. The values obtained for the inhibitory concentration of xanthyletin and avicennin are higher than those obtained for PCA therefore the inhibiting capacity is lower; The expression reported in the results: "slightly lower than PCA" refers to the values of EC50! Lower values indicate higher inhibiting capacity! It would be better to write as explained in the conclusions “xanthyletin and avicennin showed slightly lower antifungal activities against P. oryae than PCA (29.30±1.89 mg/L), with the EC50 values of 31.56±1.86 mg/L and 35.89±1.64 mg/L, respectively”.

Materials and methods section:

In the materials and methods there is no paragraph with statistical analysis performed. “All fungicidal activities were evaluated by statistical analysis” : which test was used?

Author Response

ref 12 (Sandjo, L. P., V. Kuete, R. S. Tchangna, T. Efferth and B. T. Ngadjui. 2014. Cytotoxic benzophenanthridine 221 and furoquinoline alkaloids from Zarrthoxylum buesgenii (Rutaceae). Chem. Cent. J. 8:1-5.) is not present in the introduction.

Response: Thank you for your good suggestions. The corresponding revision has been done and were highlighted using yellow color for your checking.

Results and Discussion section

Insert an explanation in the text to choose the concentrations of 500 mg / L (used for the crude extract) and 50 mg / L (as a screening concentration for each pure compound).

Response: Thank you for your good suggestions. The corresponding revision has been done and were highlighted using yellow color. Please see Page 5, line 5 in MS.

Table 3 remains unclear and the text still indicates that “fractions K against P. oryae showed moderate antifungal activities …”

Response: Thank you for your good suggestions. We are very sorry for our carelessness and the misplacement of the table data. We reedited the article carefully. The corresponding revision has been done and were highlighted using yellow color.

I know perfectly the meaning of the EC50, so it seems to me that the explanation of the results of tab 5 is not correctly expressed. The values obtained for the inhibitory concentration of xanthyletin and avicennin are higher than those obtained for PCA therefore the inhibiting capacity is lower; The expression reported in the results: "slightly lower than PCA" refers to the values of EC50! Lower values indicate higher inhibiting capacity! It would be better to write as explained in the conclusions “xanthyletin and avicennin showed slightly lower antifungal activities against P. oryae than PCA (29.30±1.89 mg/L), with the EC50 values of 31.56±1.86 mg/L and 35.89±1.64 mg/L, respectively”.

Response: Thank you for your good and valuable suggestions. We are very sorry! Maybe our last reply was not properly explained. We never meant to offend you. I certainly know and agree that you are an expert in this area. The corresponding revision has been done as followed and were highlighted using yellow color in MS.------“The results showed that the antifungal activities of xanthyletin (1) and lcuvangetin (2) against P. oryae slightly lower than PCA (29.30±1.89 mg/L), with the EC50 values of 31.56±1.86 mg/L and 35.89±1.64 mg/L, respectively. Noteworthily, avicennin (3) showed higher antifungal activity against F. graminearum with the EC50 values of 43.26±1.76 mg/L, than PCA (52.34±1.53 mg/L). The results revealed that xanthyletin, luvangetin and avicennin were the main antifungal substances of Z. avicennae.

Materials and methods section:

In the materials and methods there is no paragraph with statistical analysis performed. “All fungicidal activities were evaluated by statistical analysis” : which test was used?

Response: Thank you for your valuable suggestions. All statistical analysis was performed using EXCEL 2010 software. The log dose-response curves allowed determination of the EC50 for the fungi bioassay according to probit analysis. The 95% confidence limits for the range of EC50 values were determined by the least-square regression analysis of the relative growth rate (% control) against the logarithm of the compound concentration. The corresponding revision has been done and were highlighted using yellow color.